# Reproductive healthcare in prison: A qualitative study of women's experiences and perspectives in Ontario, Canada

Jessica Liauw[1]*, Jessica Jurgutis[2,3], Elysée Nouvet[4], Brigid Dineley[1], Hannah Kearney[5ᴑ], Naomi Reaka[5ᴑ], Donna Fitzpatrick-Lewis[6‡], Leslea Peirson[7‡], Fiona Kouyoumdjian[8]

1 Department of Obstetrics and Gynecology, University of British Columbia, Vancouver, Canada, 2 Department of Indigenous Learning, Lakehead University, Ontario, Canada, 3 Department of Women's Studies, Lakehead University, Ontario, Canada, 4 School of Health Studies, University of Western Ontario, Ontario, Canada, 5 Faculty of Health Sciences, School of Medicine, McMaster University, Ontario, Canada, 6 School of Nursing, McMaster University, Ontario, Canada, 7 Independent Researcher, Ontario, Canada, 8 Department of Family Medicine, McMaster University, Ontario, Canada

ᴑ These authors contributed equally to this work.
‡ These authors also contributed equally to this work.
* Jessica.liauw@medportal.ca

**Data Availability Statement:** All relevant data are within the manuscript and its Supporting Information files.

## Abstract

### Objective

To explore women's experiences and perspectives of reproductive healthcare in prison.

### Methods

We conducted a qualitative study using semi-structured focus groups in 2018 with women in a provincial prison in Ontario, Canada. We asked participants about their experiences and perspectives of pregnancy and contraception related to healthcare in prison. We used a combination of deductive and inductive content analysis to categorize data. A concept map was generated using a reproductive justice framework.

### Results

The data reflected three components of a reproductive justice framework: 1) women have limited access to healthcare in prison, 2) reproductive safety and dignity influence attitudes toward pregnancy and contraception, and 3) women in prison want better reproductive healthcare. Discrimination and stigma were commonly invoked throughout women's experiences in seeking reproductive healthcare.

### Conclusions

Improving reproductive healthcare for women in prison is crucial to promoting reproductive justice in this population. Efforts to increase access to comprehensive, responsive, and timely reproductive healthcare should be informed by the needs and desires of women in

**Funding:** This work was funded by an award (JL) from the Regional Medical Associates Research Scholarship Fund, in Hamilton, Ontario. The funders had no role in study design, data collection and analysis, decision to publish, or preparation of the manuscript. One author (LP) is currently employed by a public health unit but did not participate in this project in this role, so we did not list this organization as their affiliation. This author was not paid to do any of the work involved in this project by any employer (past or present) nor did they receive any of the project's funding in return for their services.

**Competing interests:** The authors have declared that no competing interests exist. One author (LP) is currently employed by a public health unit but did not participate in this study in this capacity, and this does not alter our adherence to PLOS ONE policies on sharing data and materials.

prison and should actively seek to reduce their experience of discrimination and stigma in this context.

## Background

Women experiencing imprisonment in North America lack access to reproductive healthcare [1–3]. Studies in two prisons in Canada and the US, respectively, both found that approximately 80% of women in prison had an unmet need for contraception [1,4], and internationally, women who experience imprisonment are found to have less access to prenatal care and higher rates of pregnancy complications, such as preterm birth and low birth weight, compared with women in the general population [5–9].

While previous quantitative studies from the US have shown that most women are interested in accessing reproductive healthcare while in prison [2,10–12], issues such as stigma and healthcare quality may prevent women from accessing desired care even when available [10,13]. The World Health Organization and the United Nations recommend the provision of reproductive healthcare, specifically including contraception and pregnancy-related care, for women in prison [14]. Access to contraception in prison can support women in preventing unintended pregnancy after release, which is particularly important since women may be at increased risk of unintended pregnancy during that period and they often face barriers to contraception access in the community, including urgent competing priorities while transitioning back to the community [15]. Contraception access in prison is also important for females who are sexually active with males in prison. A few qualitative studies from the US have described women's experiences and preferences regarding contraception during and following imprisonment [10,16,17], but to our knowledge there are no studies that have addressed reproductive healthcare more broadly, that is, examining perspectives and experiences of both contraception and pregnancy. Understanding women's experiences in this wider context is important to improving health outcomes and promoting reproductive justice [18], which includes 1) the right not to have a child, 2) the right to have a child, and 3) the right to parent children in safe and healthy environments.

Our objective was to explore women's experiences and perspectives of pregnancy, contraception, and related healthcare in prison.

## Methods

### Overall approach

For this qualitative study we conducted focus groups and analyzed data using a combination of deductive and inductive content analysis [19]. This approach was taken because we did not assume a pre-specified theoretical framework about women's experiences and perspectives on our issues of interest. We took a factist standpoint (assuming data to be accurate representations of reality) [20], focusing on manifest content of the data (i.e., describing what was said, rather than interpreting what was said and what was not said such as sighs, posture, laughter, etc.) [19–21].

### Study context

We conducted this study in a provincial prison in Ontario, Canada. In Canada, provincial prisons hold people admitted to custody prior to trial and people who receive a sentence of fewer

than two years in custody [22]. Provincial prisons are publicly funded and administered by the Ministry of the Solicitor General. In this paper, we use the term provincial prisons to refer to all provincial correctional facilities, and the term imprisonment to include detention (i.e., pre-trial) and incarceration (i.e., post-sentencing).

In Ontario, hospitalizations and medically necessary physician services are paid for through the public health insurance system, including in provincial prisons. In provincial prisons, pre-scribed medications are paid for by the Ministry of the Solicitor General. In the community, prescribed medications are not universally paid for, but some people, including those who receive benefits based on financial need and employment status or disability, have publicly funded coverage for prescribed medications through the Ontario Drug Benefit program [23].

We use the term women for people identifying with that gender, regardless of sex. We use the term females if cited work specified females as the population of interest.

## Study development

In 2016, we completed a survey of women in an Ontario provincial prison to quantify their unmet need for contraception [4]. We originally planned this survey as part of a mixed-methods study that included focus groups, because we anticipated that reasons for using or desiring contraception may not be fully described or understood using either a quantitative or qualitative approach independently. However, during the process of seeking approval for the protocol from the Ministry responsible for provincial prisons there were concerns raised about the burden the study would place on the institution, so we removed the qualitative component of the study. In the survey, we found that 77% of women (N = 85) had experienced an unintended pregnancy, that 80% of women who were at risk for unintended pregnancy had not been using a reliable form of contraception prior to imprisonment, and that only 44% of all participants wanted more information about contraception [4]. These results, along with written responses in which women described a wide range of salient and traumatic experiences when asked "Do you have any other comments about pregnancy or birth control?" at the end of the survey, solidified our motivation to collect qualitative data to better understand women's perspectives and experiences regarding pregnancy, contraception, and related healthcare in prison. We thought qualitative data would be valuable to inform efforts to improve reproductive healthcare in prison, including access to contraception if this emerged as being important to women. We therefore updated the protocol to conduct focus groups with women in a provincial prison. Prior to conducting the focus groups in prison, we conducted two focus groups on this topic with women who had recently been released from prison, which allowed us to pilot our focus group guide and make modifications based on this experience before starting the focus groups in prison.

## Study procedures

We planned to conduct four to six, one-hour, focus groups with four to eight women per group to explore diverse perspectives and achieve data saturation [24]. The focus group guide (S1 File) was developed by project team members (JL, JJ, EN, FK) based on the study objectives, issues identified in previous research [1,13,25], and data from our aforementioned survey [4]. The guide included questions on attitudes toward pregnancy and contraception, experiences of healthcare related to these issues, barriers to contraception, and suggestions for improving access to contraception in prison. In the focus groups, we used the term "birth control" to discuss all forms of contraception (e.g., barrier methods, oral contraceptive, intrauterine device (IUD), or other methods to prevent pregnancy).

We recruited participants through posters and announcements in the prison common areas (announcements were made by the prison's social worker) inviting women of reproductive age (specified as 18–49 years old, since only women 18 years and older were admitted to that facility) to participate in a study focusing on birth control, pregnancy, and reproductive health. We included English-speaking women who were able to provide voluntary informed consent.

The focus groups were facilitated by one female team member (JJ), who had graduate-level training in qualitative methods and feminist epistemologies. The facilitator did not have a prior relationship with any study participant. Only the participants and facilitator were present for focus groups. The facilitator met each group of interested women at a scheduled time. She obtained written consent for participation and audio recording of the focus group. In the letter of information participants were informed that they were able to opt out of the study at any time, and that participation in the study would not impact their treatment in the facility. They were asked to be respectful of each other's experiences given the sensitive and personal nature of the conversations and to keep what was shared in the group confidential and were also reminded that confidentiality could not be guaranteed following the group. The facilitator reviewed the letter of information and consent form verbally and also made participants aware of her affiliation, research background, context of the current research as situated in the previous research completed on the topic by the researchers, and the objectives of the research. The facilitator led semi-structured discussions using the focus group guide. The facilitator made field notes following each focus group, however they were for the facilitator's personal use and were not included in the analysis. No compensation was provided to participants, consistent with the Ministry of the Solicitor General policy. Transcripts were not returned to participants for comment.

## Analysis

We transcribed focus group recordings verbatim. We conducted deductive and inductive content analysis [19] of the focus group data using NVivo software. In the preparation phase, four project team members (JL, JJ, BD, FK) read all the transcripts to become familiar with the content. All four team members then coded one transcript according to an unconstrained categorization matrix based on the questions in the focus group guide (S1 File), and the matrix was adjusted after group discussion. Two team members then independently coded each remaining transcript using the finalized matrix. Because the matrix was unconstrained, subcategories were inductively created within each category [19]. We compared coding results and resolved differences by consensus. After initial review of our results, we considered using a reproductive justice framework to enhance our analysis and interpretation [18]. Results were not returned to participants for checking.

The study was approved by the Hamilton Integrated Research Ethics Board (13–614) and the Ontario Ministry of the Solicitor General.

## Results

We conducted three focus groups with seven women in each group, for a total of 21 participants in a provincial prison. The focus groups occurred in August 2018 and lasted between 60 and 90 minutes. Women were between the ages of 20 and 44 years. No participants dropped out of the study once written consent was obtained.

Our initial coding matrix included the following categories: 1) experiences and attitudes about health and reproductive health, 2) experiences and attitudes toward pregnancy, 3) experiences and attitudes toward contraception, 4) barriers to accessing contraception, and 5)

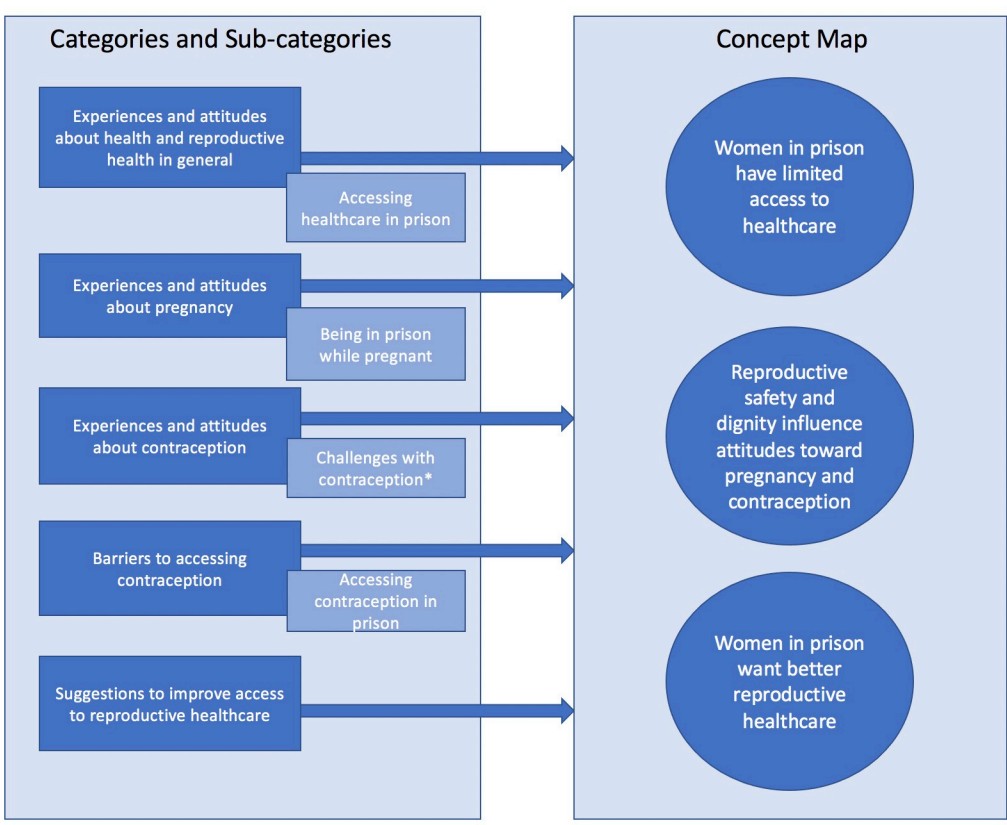

**Fig 1. Summary of categories and reproductive justice concept map.** Categories derived from the focus group guide, sub-categories derived inductively and concept map interpreted using a reproductive justice framework. *Sub-category 'challenges with contraception' was not included in our final analysis in order to streamline results, as the relevant concepts within this sub-category were captured within the other categories.

suggestions to improve access to reproductive healthcare. Within these categories, the data suggested four sub-categories (see Fig 1). Overall, when asking women in prison about their experiences and perspectives on pregnancy, contraception, and reproductive healthcare, women discussed concepts reflecting the right to have children, the right not to have children, and the right to parent the children they have in safe and sustainable environments–that is, the three pillars of the reproductive justice framework [18]. Guided by this framework, we synthesized our categories into a concept map with the following components: a) women in prison have limited access to healthcare, b) reproductive safety and dignity influence attitudes toward pregnancy and contraception, and c) women in prison want better reproductive healthcare. Saturation of data was reached within these components [24]. We omitted the sub-category of 'challenges with contraception' in our final analysis since the most salient aspects of this sub-category pertaining to reproductive justice were captured within the other categories about contraception (Fig 1).

## 1) Women in prison have limited access to healthcare

Participants described having limited access to healthcare in prison. They attributed this limitation to a lack of available healthcare personnel (e.g., nurses, physicians), materials (e.g., bandages), the hierarchical prison structure, discrimination, favouritism, and lack of trust.

Many reported waiting months to see a physician despite multiple requests. Others explained that people in prison must submit healthcare requests to correctional officers, who decide whether to submit their requests to the physician.

*There's a lot of freaking people in here, a lot of girls in here, just to have a doctor in here one day a week. She has all these appointments in the morning, she doesn't even get to see half the girls, right.*

*There's no guarantee that you're going to get that medical treatment, or that you're going to be able to talk to the doctor, or the psychiatrist, or the nurse. It's up to the [correctional officer], because they're the one that passes on that message, they're the one that puts you on the list to see the doctor.*

*When you're in jail you don't have the opportunity to make your own decisions. They make them for you. . .. With health care, they'll decide whether you need to go see someone outside as opposed to the doctor on site, for a specialist or whatever.*

Some participants believed that correctional officers did not consider them as deserving of good healthcare as members of the general population. Some felt that in prison, healthcare provided to women was worse than healthcare provided to men. Participants reported that favouritism resulted in unequal distribution of healthcare and supplies among women in prison.

*Oh, well, she was on the streets to begin with, so, who cares that we're not giving her the right amount of food? I think that's what their [correctional officers'] mindset is, 100 percent.*

*. . . [L]ike just even with methadone. The men get it before they go to court and the women, we can't have it, we don't get it.*

*. . .and some nurses will, if they like you, they'll give it [cream for a rash] to you anyway. But if, you know, if they don't really know you, they don't care.*

Many women felt that their health concerns were not taken seriously until their conditions were severe; one woman said she had to be "bleeding and crying" to get medical attention. Another said, "to go to the hospital, you need to be, like dying". Some thought that nurses generally did not respect or care about them, which was disappointing for those who considered being in prison as the "last hope" to improve their health.

Participants also described the ways in which incarceration directly limited their ability to access essential reproductive healthcare. For example, several women reported not being able to access follow-up care for IUDs, and at least two participants said they, or others they knew, experienced distress due to the inability to access abortion services or a pregnancy test in prison.

*[regarding an IUD]. . .we can't find the strings, we don't know where it is. So, then I want to go get an ultrasound done, but I came in here before [being able to] find out the results of the ultrasound by going to my doctor, so I mentioned it to the nurse here and like, who knows what's going on with the IUD up there and it's been up there like, for who knows how long and I still haven't seen anybody to do an ultrasound to checkup on me, right?*

*. . .but one of my best friends is in here and she was already three months pregnant and she was begging and begging, like she's done requests, she's seen a doctor, she's already planned to*

*go for an abortion. They were putting it off and putting it off, until like, almost at the point that she couldn't get one. Luckily she got released and she was able to go get an abortion.*

## 2) Reproductive safety and dignity influence attitudes toward pregnancy and contraception

Within the reproductive justice framework, reproductive safety and dignity depend on having broader essential needs met: housing, a living wage, the ability to live free of racism, a healthy environment, etc. (page 56) [18]. When asked about their perspectives on pregnancy, pregnancy timing, and contraception, participants discussed a range of these topics pertaining to both time in prison and outside of prison. They described that how they felt about a pregnancy would depend on factors in their lives such as stability and whether the pregnancy was planned. Participants from all focus groups described the importance of being "ready" for pregnancy: mentally, emotionally, physically, socially, and financially. Some participants shared that previous experiences of not being ready for pregnancy had led to challenges, and "heartache" after being separated from a child.

*When I had my son, I wasn't ready, so it caused a lot more heartache and pain after because I had to put him with my mom. So, it kept me going down the wrong path because I wasn't there for him and I felt bad about it, so I wasn't coping well.*

Some women desired pregnancy. One participant was concerned that being in prison during her reproductive years may mean losing the opportunity to become pregnant.

*You don't even realize how bad I wish I had gotten pregnant before I got here because I want another baby so badly. And, I'm looking at six years, so I don't even know if I will be able to have another baby when I get out, maybe.*

Participants thought it was important for women to have control over when they get pregnant but had varied opinions on whether having control was possible. Several participants shared stories of becoming pregnant unexpectedly while on birth control, or after assuming they could not get pregnant. For example, one participant had been told her ovaries were removed when she was a teenager. Other participants shared that they thought they could not get pregnant because of infrequent periods.

## 3) Women in prison want better reproductive healthcare

When we asked women about their experiences with contraception and pregnancy while in prison, they outlined the importance of improving healthcare in prison for these issues and made several pragmatic suggestions to work towards this, such as having access to a gynecologist or female-specific healthcare. Women again shared multiple examples of discrimination and stigma acting as barriers to healthcare in this context. We describe data for contraception and pregnancy separately, below.

**Contraception.** Participants discussed the importance of contraception and identified barriers to access for women in prison. Several participants noted that the time in prison would be a good time to access contraception because "people's minds are more clear [because they are not using drugs]," and "people sort of have a moment of pause". In addition, some thought it would be important to access contraception in prison because they anticipated increased fertility after a period of not using drugs. Finally, some women felt it was important

to have consistency with a given contraceptive method, so they would want to continue what they were on prior to entering prison or start a method before release so they had a chance to get used to it.

> *I think that it would be really good because, if they offered birth control, especially like, even though we're not at high-risk of pregnancy in here, women are super sexually active when they leave here, because they've been held for so long. I know so many women who leave a facility or an institution like here and get pregnant.*

However, participants had varied understanding of and experience with access to contraception in prison. Several women did not think it was possible to start contraception in prison, and one said she had asked for contraception while in prison but was declined. Another woman said that some people in prison are denied contraception because, in her understanding, some doctors do not like to prescribe it. Several participants said healthcare staff did not offer or ask about contraception.

> *We might be offered it, like they might put you on birth control here, birth control might be an option, but it's not talked about, you know, so, it's not—we're not familiar with it.*
>
> *[. . .] unless you've been taking [contraception] before you came to jail, they won't give it to you.*

Some participants described barriers to accessing condoms and dental dams while in prison. One participant said that she had to ask correctional officers for these items, rather than healthcare staff, which made her feel "uncomfortable". Some women said that nurses and correctional staff did not provide or allow people in prison to have condoms because they were considered "contraband".

> *So, we can't ask for contraception like you said, like you were asking, because we're not allowed to, . . . so, there is no contraceptive and to even ask, we heard it's on the cart, but to ask in front of a guard, that's just like saying, hey, I'm about to do a misconduct, or try to do something inappropriate.*

Across groups, participants suggested that healthcare staff should ask women about their current needs for and interest in contraception. Some participants thought contraception should be discussed and offered during the routine nursing assessment on prison admission, and that people should be made aware that contraception can be accessed at any time during imprisonment.

Participants also had ideas about increasing access to information about contraceptive methods while in prison. At least two participants suggested having posters or written material describing available options before seeing a physician, to optimize time during the appointment. Others suggested having group sessions led by a visiting public health nurse, which some participants had experienced in the past in prison and had found helpful. Participants suggested topics for discussion such as why contraception is important, who should use it, and how to access options like an IUD.

> *I think it should be, you know how we get asked our medical conditions when we come in, that form; I think in that form they should put it as, do you wish to go on birth control, . . . and then also in the same sentence, at any time you have the right to go on birth control if you choose to. Yes or no or talk about it another time.*

*Yeah. I think they should be given the option, and then the doctor should be given some kind of file folder with all the drugs. So, they could say, this is what we got, here's the information on all the pros and cons, take this back to your cell, let the nurse know what you want.*

Participants also described barriers to continuing or initiating contraception after release from prison, including not being provided with a supply of contraception on release, not being connected to healthcare, and not having the appropriate identification to access healthcare.

*When I got out from [prison], I was going to get on birth control. Every single time I got sent back, the police threw away my ID; so, every time I got out, I'd have to start over and get my birth certificate and my health card.*

Participants discussed the prohibitive costs of some types of contraception in the community, including an IUD ("several hundred dollars"), Plan B ("$50"), and latex-free condoms. One participant contrasted discharge planning with respect to contraception to the support she received around her diabetes care. For diabetes, appointments were set up and she was given maps and contact information to help her get to the appointments. She believed the level and continuity of support she received for her diabetes care should be the same for all health issues.

Participants suggested that contraception should be discussed during routine release planning, including options to access a supply of contraception before release and information about services in the community.

*They should have resources for contacts to services/programs in different communities. Like clinics if you don't have a doctor. This should be part of the general services made available to people alongside other needs like housing, etc. because things change and some people have been locked up a long time, or have other barriers, like shyness, etc.*

**Pregnancy.** Several participants shared experiences involving a lack of medical, psychological and social supports when dealing with pregnancy loss while in prison. They described having no access to sanitary pads while experiencing a miscarriage, and receiving delayed supports or a complete lack of medical or social supports.

*Like, I quietly, secretly had a miscarriage in here, and nobody helped me at all.*

*My friend, she lost her baby when she was in jail, and I find that she's not getting any emotional support for that and she's not able to talk about it. . .*

In most groups, participants shared concerns or experiences of physical violence from correctional staff and other people toward imprisoned women, including those who are pregnant. Some participants expressed that aggressive treatment or exposure to violence that they experienced at the time of their arrest or in prison may have led to miscarriage.

*When I was arrested, originally, I was pregnant and I didn't know I was pregnant, so like the officers were so aggressive with me that it caused me to have a miscarriage.*

*Like, I was at the police station and I was like bleeding like so bad, and they only gave me one pad. They only gave me one pad the whole time, and my pants, I had blood everywhere like,*

*then I like, it took them hours before they called an ambulance and took me to the hospital, and then they found out I miscarried. So, that was kind of traumatizing, you know.*

Most participants said that being pregnant while in prison would be stressful. Many were concerned about the inability to access healthcare or medications like prenatal vitamins, to have a healthy lifestyle, and to be in a healthy environment during pregnancy. For example, participants explained that in prison it would take weeks to get a healthcare appointment and that women are not informed of the results of their blood tests or ultrasounds. Participants in all focus groups discussed challenges getting adequate and nutritious food while pregnant in prison. Some said pregnant women had access to more food than other people in prison, and others said these extra portions had decreased in quantity and quality over time. They also discussed other difficulties of being in prison during pregnancy, such as getting adequate sleep, exposure to loud and disorienting noises, exposure to stress that they worried may affect the baby, lack of exercise, lack of sanitary facilities including living quarters, and lack of emotional support as they were separated from their partner.

*It takes a long time to get anything done. Like if I'm having symptoms now, today being pregnant and I tell the doctor, I'm not going to see her for another two to three weeks. So, anything that's going on with my baby, like my baby could be dead inside of me for two weeks before. . .*

*They [correctional officers] even accused the woman that was pregnant of going out and trying to steal her own pregnant vitamin. . .They're like, oh, you don't look pregnant, there's other women here that look more pregnant than you. Like they didn't think the vitamin she was trying to go up and get was even hers. . .I would be too scared to get pregnant here.*

*. . .to the correctional officers here, you're just another inmate. They don't care that you're in jail and you're pregnant. You're in jail. It's your fault that you're here, whether you're pregnant or not. They treat you just like everybody else, regardless of what your specific needs are.*

Many participants also expressed concern about the separation of women and their babies after delivery, some describing this as "horrific".

*Yeah. It's the first thing that runs through every woman's mind in jail is, they're gonna take my baby away from me.*

Participants also gave several examples of supporting others who were pregnant in prison. These included saving portions of their meals for pregnant women, providing emotional support during pregnancy and after pregnancy loss, and providing support and advocacy during labour.

*I was freaking out, because I had to stay and take care of her. It happened at night time. So, when the guards would leave, I was sitting there rubbing her feet, rubbing her back, you know, getting a water bottle, like hot water, the soap bottles and stuff like that; and a lot of the other girls were already sleeping because this happened in the middle of the night. . .. They keep saying, count her contractions, she's not supposed to. She has to go. I said, her date is [date specified] to get a C-section. It's written down, you know.*

Two participants expressed that compared with homelessness or living in a shelter while pregnant, imprisonment meant better access to food and a reliable place to sleep.

## Discussion

Study participants identified multiple barriers to general healthcare and reproductive healthcare in prison. They had trouble initiating, discontinuing, and following up with contraception; and with addressing pregnancy-related needs regarding miscarriage, abortion, antenatal care, labour and delivery, and postpartum care. In particular, with respect to pregnancy, participants shared how their lack of access to adequate healthcare and health resources contributed to trauma surrounding pregnancy loss or the potential for pregnancy loss while being detained. Participants also described other factors contributing to lack of access to healthcare and reproductive healthcare, including discrimination, favouritism, prison conditions, hierarchies and abuses of power, and experiences of violence. Participants highlighted the importance of having other essential socio-economic and health needs met in order to be ready to have a child, or in order to control not to have a child. Many wanted improved reproductive health services and had suggestions about how to achieve this.

The results of this study are consistent with prior research. US studies involving women in prison [1–3,12,13,25–27] have also identified barriers to accessing contraception in prison and after release, despite women's interest in and motivation to access contraception [1,13,28]. A small number of studies have used qualitative approaches to explore reproductive health issues among women in prison [10,16,17,29]; these studies examined specific issues such as sterilization [16] and the prevention of sexually transmitted infections on release [29]. US women interviewed for one study thought contraception services should be available and had concerns about the quality of care in prison and about community follow-up, and some reported the desire to become pregnant [10]. Quantitative studies from the US and Canada also indicate that women experiencing imprisonment have less access to pregnancy-related care compared with non-incarcerated populations [8,30], and a survey of prison wardens in the US found that pregnant women in prison had unmet needs regarding nutrition, rest, and psychosocial support [9]. Our finding that women in prison are made to feel like their health concerns are not legitimate has been echoed in other work [31,32]. Previous studies have identified that some women see their time in prison as an opportunity to access healthcare [31,33], including contraceptive services [13]. A recent Canadian study conducted focus groups with 11 admitted women and six healthcare staff in a provincial correctional facility, and similar to our study, found that factors influencing the use of women's health services in prison were lack of gender-specific services, mistrust of healthcare providers, and fragmentation of healthcare [33]. Our study expands on this research and adds critical insights regarding the potential impact of discrimination and stigma as direct barriers to reproductive healthcare. By examining both contraception and pregnancy 'side-by-side,' the underlying and systemic issues which inhibit access to reproductive healthcare and reproductive justice for this population were further elucidated.

Our study has some limitations. First, we did not ask women directly about experiences with and perspectives on abortion, or abortion care in prison, which is an aspect of reproductive justice [18]. We used the term 'women' to recruit participants, and people self-selected into this category. We acknowledge that experiences of pregnancy and reproductive health care are not confined to those who identify as women; however, we did not specifically recruit trans-identified or gender non-conforming persons for this study. It is possible that some people who have accessed, or who desired access to reproductive healthcare in prison were excluded from our study on the basis of their gender identity. Since it is well documented that trans and gender-non-conforming persons experience greater barriers to healthcare in prison and in the community, we believe it is important for future studies to explore these experiences by recruiting participants based on experiences of reproductive healthcare, rather than the

category of 'women.' Participants were recruited from only one provincial prison, however, the data reached saturation within the components of the reproductive justice concept map. The primary investigator of this study (FK) worked as a family physician in the prison where groups were conducted, as was noted on the information and consent forms, which may have affected participation and discussion. In addition, our use of focus groups may have prevented some personal stories from being shared by some of the women; however, this data collection strategy may have also promoted richer discussions of shared experiences [34].

The experiences of participants in this study exemplify and are consistent with ways in which imprisonment interferes directly with reproductive justice, i.e., the ability to decide if and under which conditions a woman will or will not have a baby [35]. Improving access to reproductive healthcare is a clear way to promote reproductive justice for women in prison, especially because it is well-documented that time in prison can serve as an opportunity to address healthcare needs [1,4,13,15,26,36]. Further, as detailed in the *United Nations Rules for the Treatment of Women Prisoners and Non-custodial Measures for Women Offenders* (Bangkok Rules), women in prison have a right to accessible healthcare, a healthy environment, and access to reproductive and sexual health services [37]. Reproductive justice is, however, more than access to healthcare–it is the "splicing together the equation of reproductive rights plus social justice" (page 65) [18]. While previous publications have suggested policies and healthcare practices which might improve reproductive healthcare [38,39], our findings support the fact that reproductive justice in the prison setting will only be fully achieved when discrimination, hierarchies of power, and stigma against women in prison are addressed as barriers to health. Our results also raise the question, as posed by other reproductive justice scholars [40], as to whether reproductive justice can in fact be achieved in prison since these data suggest that women's desires to have better quality and access to reproductive healthcare, and healthcare overall, has largely been negatively impacted by the conditions of their arrest and incarceration. Ultimately, although our study was focused on reproductive healthcare, our findings highlight ways in which systemic changes are necessary to promote the overall health of women in prison.

Our findings can inform interventions to improve the quality of reproductive healthcare in prison in ways that "disrupt the dehumanizing status quo of reproductive politics" and promote reproductive justice, which includes equity, freedom and dignity (page 11) [40]. Participants made several suggestions about ways to improve reproductive healthcare in prison, including making information regarding contraceptive options more readily available prior to seeing a doctor, asking women about contraceptive preferences on admission to the prison, and improving continuity with services in the community to access contraception. Future programs for women in prison should actively work to eliminate discrimination, stigma, and intersecting oppressions that women experiencing imprisonment may face. Given the long history of coercive and inhumane reproductive treatments of marginalized and imprisoned women in North America, particularly as experienced by Black and Indigenous women [41–43], it is of particular importance that interventions center the experiences of and be informed by the needs and desires of imprisoned and formerly imprisoned women. Interventions should not pathologize or stigmatize experiences and choices surrounding contraception or pregnancy in marginalized populations, and should take a patient-centered approach in their development and implementation, while contributing to the prevention of imprisonment through efforts that support decriminalization and decarceration [40,42,44].

## Supporting information

**S1 File. Focus group guide.**
(DOCX)

## Author Contributions

**Conceptualization:** Jessica Liauw, Donna Fitzpatrick-Lewis, Leslea Peirson, Fiona Kouyoumdjian.

**Formal analysis:** Jessica Liauw, Jessica Jurgutis, Brigid Dineley, Hannah Kearney, Naomi Reaka, Fiona Kouyoumdjian.

**Funding acquisition:** Jessica Liauw, Fiona Kouyoumdjian.

**Investigation:** Jessica Liauw, Jessica Jurgutis, Elysée Nouvet, Fiona Kouyoumdjian.

**Methodology:** Jessica Liauw, Jessica Jurgutis, Elysée Nouvet, Donna Fitzpatrick-Lewis, Leslea Peirson, Fiona Kouyoumdjian.

**Project administration:** Jessica Liauw, Fiona Kouyoumdjian.

**Supervision:** Fiona Kouyoumdjian.

**Writing – original draft:** Jessica Liauw.

**Writing – review & editing:** Jessica Jurgutis, Elysée Nouvet, Brigid Dineley, Hannah Kearney, Naomi Reaka, Donna Fitzpatrick-Lewis, Leslea Peirson, Fiona Kouyoumdjian.

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
