## [Decision Letter · Decision Letter 0]

31 Dec 2020

PONE-D-20-34245

Contraception and reproductive healthcare in prison: A qualitative study of women’s experiences in Ontario, Canada

PLOS ONE

Dear Dr. Liauw,

Thank you for submitting your manuscript to PLOS ONE. After careful consideration, we feel that it has merit but does not fully meet PLOS ONE’s publication criteria as it currently stands. Therefore, we invite you to submit a revised version of the manuscript that addresses the points raised during the review process.

I am enthusiastic about the data that was collected, but share the concerns of the reviewers regarding the superficial analysis and presentation of the results. While incorporating a more robust theoretical framing holds the potential to improve the manuscript greatly, a more in-depth analysis, clearer reporting of the methods and nuanced discussion of the findings are needed. This is not a guarantee of publication after revision, but I look forward to seeing a revised manuscript that meaningfully addresses all of the concerns outlined by the reviewers below. 

We look forward to receiving your revised manuscript.

Kind regards,

Andrea Knittel

Academic Editor

PLOS ONE

Additional Editor Comments:

As I have noted in my letter above, I am in agreement with the reviewers that a lack of theoretical grounding and careful analysis may be difficult to overcome. However, the data are interesting and the topic is important. I have outlined my major concerns below.

1. Please submit a completed COREQ checklist with the revised manuscript. Ensuring compliance with the Equator Network guidelines helps with methodological rigor and reporting.

2. Consider moving beyond a summative approach to the qualitative data to a more theoretically driven analysis. As the reviewers note, a reproductive justice framework might be a useful framing for the paper, although there are other frameworks that would help to guide interpretation of the data. I also agree with reviewer #2 who suggested that presenting the quotes in tables makes it difficult to read and connect with the text.

3. Given the extended time period and multiple settings, the methods section should include some justification for analyzing these data together. Were the two sets of qualitative data analyzed separately, and similar themes were encountered? Was saturation achieved? Being explicit in the analysis about which themes were identified by the participants residing in the community and which by currently incarcerated participants may help with this as well.

4. The manuscript is somewhat disjointed, which I attribute to a lack of theoretical underpinning. Framing from the introduction in a theoretically motivated way will allow for carry through into the analysis and reporting of the results and discussion of the findings.

Journal Requirements:

2. Please include a copy of the interview guide used in the study, in both the original language and English, as Supporting Information, or include a citation if it has been published previously.

3. Please provide additional information regarding the considerations  made for the prisoners included in this study. For instance, please discuss whether participants were able to opt out of the study and whether individuals who did not participate receive the same treatment offered to participants.

We note that one or more of the authors are employed by a commercial company:Independent Researcher.

4.1. Please provide an amended Funding Statement declaring this commercial affiliation, as well as a statement regarding the Role of Funders in your study. If the funding organization did not play a role in the study design, data collection and analysis, decision to publish, or preparation of the manuscript and only provided financial support in the form of authors' salaries and/or research materials, please review your statements relating to the author contributions, and ensure you have specifically and accurately indicated the role(s) that these authors had in your study. You can update author roles in the Author Contributions section of the online submission form.

4.2. Please also provide an updated Competing Interests Statement declaring this commercial affiliation along with any other relevant declarations relating to employment, consultancy, patents, products in development, or marketed products, etc.  

Reviewers' comments:

Reviewer's Responses to Questions

**Comments to the Author**

1. Is the manuscript technically sound, and do the data support the conclusions?

Reviewer #1: Yes

Reviewer #2: No

Reviewer #3: Yes

2. Has the statistical analysis been performed appropriately and rigorously? 

Reviewer #1: N/A

Reviewer #2: N/A

Reviewer #3: N/A

3. Have the authors made all data underlying the findings in their manuscript fully available?

Reviewer #1: Yes

Reviewer #2: Yes

Reviewer #3: Yes

4. Is the manuscript presented in an intelligible fashion and written in standard English?

Reviewer #1: Yes

Reviewer #2: Yes

Reviewer #3: Yes

5. Review Comments to the Author

Reviewer #1: Thank you for the opportunity to review this manuscript on an important topic. Your background on the provincial prisons is helpful and contributes to our understanding of the context. I have two major and two minor comments.

A minor comment on page 6 is about including women aged 15 to 49 years. Can you clarify if juveniles are incarcerated in the provincial prisons—it would seem that women 15-17 years old would be in juvenile facilities. Perhaps include this in the description of provincial prisons.

Page 7, line 147 needs a citation.

One major concern is that lack of any type of social or reproductive justice framework in which to analyze and apply findings. If the authors might consider retrospectively applying one, I think it would address both of the below concerns

Of the five themes that were identified one seems out of place. From the title, “Contraception and reproductive healthcare in prison”, the theme "the importance of contraception and barriers to access for women in prison” and its related quotes seem out of place. The quotes are not about health care, they are about how women in the target population feel about contraception; this would seem to be a separate topic. I found the theme and quotes distracting and out of sync with the rest of the themes and suggest deleting.

On page 23, I believe that a final statement is weak, “Our findings can inform interventions to improve the quality and accessibility of healthcare in prison and post-release and support the development of programs that make essential services such as contraception a routine part of the care of women in prison.” The quotes in the manuscript are so disturbing, surely we want to do more with them than “inform interventions”. I do not mean to impose on own agenda on the authors, but suggest that the reason we do this type of research is to use the voices of women in this population to make changes. A stronger final statement would be appropriate.

Reviewer #2: Abstract: Too general e.g., 'structural and cultural barriers.' Share specific findings from paper.

Introduction: Need to better define the frame of reference for this study. It starts with women worldwide, but most of the cites about access to services appear to refer to studies in North America. Later, the rationale is presented that it is important to know about women in Canada because there is a national health system. In this case, the background presented should refer to incarcerated women in Canada or incarcerated women in countries with national health systems.

Methods: Sample is weak - women recruited from two different types of venues two years apart. Qualitative analysis methods inadequately described.

Results: results are organized by topic area, but not by theme as the authors state. For example, gatekeeping by prison personnel would be a theme. "Perspectives on pregnancy" is a topic. There is very little thematic analysis in this paper.

Results shift back and forth between what the women experience in prison and what the women experience more generally. This weakens the uniqueness of the paper, which is what happens around reproductive health while IN prison.

Quotes should be integrated in text near the findings you describe, not grouped together in boxes. “Illustrative quotes” is not a term typically used in qualitative analysis. Some of the quotes don’t link up with any of the findings you discuss.

The paper is lacking in analysis and insight. The conclusion, that incarcerated women should be given access to reproductive health care, is true but not unique and not directly tied to most of the findings that the authors present.

Reviewer #3: The article is an important contribution to the literature on incarcerated women's health and access to reproductive healthcare during imprisonment. There are two areas where this article can be strengthened and improved. One, the authors neglect to cite literature on family planning for incarcerated and formerly incarcerated women. Doing so would further substantiate some of the findings in this study. Second, the authors should also address potential ethical issues that are relevant when conducting studies in prison and how they addressed those ethical issues in their work. For example, receiving IRB approval is the bare minimum standard. What ethical issues did they consider prior to launching their work and how did they resolve them to proceed with this work.

6. PLOS authors have the option to publish the peer review history of their article (what does this mean?). If published, this will include your full peer review and any attached files.

Reviewer #1: No

Reviewer #2: No

Reviewer #3: No

---

## [Author Response · Author response to Decision Letter 0]

13 Feb 2021

EDITOR COMMENTS:

1. Please submit a completed COREQ checklist with the revised manuscript. Ensuring compliance with the Equator Network guidelines helps with methodological rigor and reporting.

Response: A COREQ checklist has been submitted with the revision.

2. Consider moving beyond a summative approach to the qualitative data to a more theoretically driven analysis. As the reviewers note, a reproductive justice framework might be a useful framing for the paper, although there are other frameworks that would help to guide interpretation of the data. I also agree with reviewer #2 who suggested that presenting the quotes in tables makes it difficult to read and connect with the text.

Response: We have revised and extended our analysis using a deductive content analysis approach and a reproductive justice framework (see the new ‘analysis’ section within Methods). We have added the quotes, and have removed the tables, into the text to facilitate connection. We thank the reviewers for these suggestions which we feel have strengthened the paper.

3. Given the extended time period and multiple settings, the methods section should include some justification for analyzing these data together. Were the two sets of qualitative data analyzed separately, and similar themes were encountered? Was saturation achieved? Being explicit in the analysis about which themes were identified by the participants residing in the community and which by currently incarcerated participants may help with this as well.

Response: Thank you for this comment. We have removed the data from the community-based focus groups to improve clarity. The paper now reports data from the in-prison focus groups only.

4. The manuscript is somewhat disjointed, which I attribute to a lack of theoretical underpinning. Framing from the introduction in a theoretically motivated way will allow for carry through into the analysis and reporting of the results and discussion of the findings.

Response: We have modified all sections of the paper to be more cohesive around a reproductive justice framework, as described in the Analysis section of the Methods.

JOURNAL REQUIREMENTS

Response: Thank you. We have adjusted the manuscript to meet the style requirements.

2. Please include a copy of the interview guide used in the study, in both the original language and English, as Supporting Information, or include a citation if it has been published previously.

Response: We have added the focus group guide as supporting information.

3. Please provide additional information regarding the considerations made for the prisoners included in this study. For instance, please discuss whether participants were able to opt out of the study and whether individuals who did not participate receive the same treatment offered to participants.

Response: We have added the following text “Participants were informed that they were able to opt out of the study at any time, and that participation in the study would not impact their treatment in the facility.” (line 148-149).

We note that one or more of the authors are employed by a commercial company: Independent Researcher.

4.1. Please provide an amended Funding Statement declaring this commercial affiliation, as well as a statement regarding the Role of Funders in your study. If the funding organization did not play a role in the study design, data collection and analysis, decision to publish, or preparation of the manuscript and only provided financial support in the form of authors' salaries and/or research materials, please review your statements relating to the author contributions, and ensure you have specifically and accurately indicated the role(s) that these authors had in your study. You can update author roles in the Author Contributions section of the online submission form.

Response: The author listed as ‘Independent Researcher’ is currently employed by a public health unit but did not participate in this project in this role, so we did not list this organization as their affiliation. This author was not paid to do any of the work involved in this project by any employer (past or present) nor did they receive any of the project’s funding in return for their services. We have clarified this in the funding statement (see cover letter). Please let us know if it would preferable to list the author’s location (e.g., Dundas, Ontario), as opposed to an affiliation (i.e. ‘Independent Research’) in this circumstance.

4.2. Please also provide an updated Competing Interests Statement declaring this commercial affiliation along with any other relevant declarations relating to employment, consultancy, patents, products in development, or marketed products, etc. 

Response: Thank you. We have updated the Funding Statement and Competing Interests Statement in our cover letter.

Response: This has been added.

REVIEWER COMMENTS

Reviewer #1: Thank you for the opportunity to review this manuscript on an important topic. Your background on the provincial prisons is helpful and contributes to our understanding of the context. I have two major and two minor comments.

A minor comment on page 6 is about including women aged 15 to 49 years. Can you clarify if juveniles are incarcerated in the provincial prisons—it would seem that women 15-17 years old would be in juvenile facilities. Perhaps include this in the description of provincial prisons.

Response: We have corrected our original error. While earlier versions of our protocol included 15-49 years old, our final inclusion criteria used in recruitment materials, and the minimum age range for women admitted to the facility, was 18 years old (line 135-136).

Page 7, line 147 needs a citation.

Response: We have clarified the description of our approach and have added a reference (Elo S, Kyngas H. The qualitative content analysis process. J Adv Nurs 2008; 62(1): 107-15, specified in ‘Overall Approach’ and ‘Analysis’ (line 69, 159, respectively).

One major concern is that lack of any type of social or reproductive justice framework in which to analyze and apply findings. If the authors might consider retrospectively applying one, I think it would address both of the below concerns

Response: Thank you for this suggestion. We have applied a reproductive justice framework, as described in the Analysis section of the Methods. 

Of the five themes that were identified one seems out of place. From the title, “Contraception and reproductive healthcare in prison”, the theme "the importance of contraception and barriers to access for women in prison” and its related quotes seem out of place. The quotes are not about health care, they are about how women in the target population feel about contraception; this would seem to be a separate topic. I found the theme and quotes distracting and out of sync with the rest of the themes and suggest deleting.

Response: We agree with the reviewer, and have deleted the content and quotes dealing with how women in the target population feel about and challenges with specific types of contraception. This is described in ‘Results’ (lines 194-196).

On page 23, I believe that a final statement is weak, “Our findings can inform interventions to improve the quality and accessibility of healthcare in prison and post-release and support the development of programs that make essential services such as contraception a routine part of the care of women in prison.” The quotes in the manuscript are so disturbing, surely we want to do more with them than “inform interventions”. I do not mean to impose on own agenda on the authors, but suggest that the reason we do this type of research is to use the voices of women in this population to make changes. A stronger final statement would be appropriate.

Response: We agree and have changed the sentences to: Our findings can inform interventions to improve the quality of reproductive healthcare in prison in ways that “disrupt the dehumanizing status quo of reproductive politics” and promote reproductive justice, which includes equity, freedom and dignity (page 11) [40]. (lines 516-518).

Reviewer #2: Abstract: Too general e.g., 'structural and cultural barriers.' Share specific findings from paper.

Response: We have modified the abstract to include more specific results: “The data reflected three components of a reproductive justice framework: 1) women have limited access to healthcare in prison, 2) reproductive safety and dignity influence attitudes toward pregnancy and contraception, and 3) women in prison want better reproductive healthcare. Discrimination and stigma were commonly invoked throughout women’s experiences in seeking reproductive healthcare.” (lines 32-36).

Introduction: Need to better define the frame of reference for this study. It starts with women worldwide, but most of the cites about access to services appear to refer to studies in North America. Later, the rationale is presented that it is important to know about women in Canada because there is a national health system. In this case, the background presented should refer to incarcerated women in Canada or incarcerated women in countries with national health systems.

Response: We have modified our introduction (i.e. ‘Background’) in keeping with reviewer suggestions to incorporate a reproductive justice framework. In doing so, we have referenced evidence from multiple jurisdictions (e.g., Canada, Australia, and the United States), since we believe this evidence is at least partly generalizable to our target population, specifically when considering a reproductive justice framework.

Methods: Sample is weak - women recruited from two different types of venues two years apart. Qualitative analysis methods inadequately described.

Response: We have removed the community-based sample, and now only include focus groups conducted with women in prison in 2018. We have improved the description of our qualitative methods, specifying a combined deductive and inductive content analysis approach focused on a factist philosophy and the manifest data content (lines 68-74, and 158-173).

Results: results are organized by topic area, but not by theme as the authors state. For example, gatekeeping by prison personnel would be a theme. "Perspectives on pregnancy" is a topic. There is very little thematic analysis in this paper.

Response: We have clarified that we report categories of data which were synthesized in a concept map (Figure 1), in keeping with content analysis methodology. We have reorganized and modified the results section accordingly.

Results shift back and forth between what the women experience in prison and what the women experience more generally. This weakens the uniqueness of the paper, which is what happens around reproductive health while IN prison.

Response: We agree, and have decided to focus on what women experience in prison, and the content which pertains to a reproductive justice framework. We have removed the data from the focus groups conducted in the community, and have modified the results accordingly.

Quotes should be integrated in text near the findings you describe, not grouped together in boxes. “Illustrative quotes” is not a term typically used in qualitative analysis. Some of the quotes don’t link up with any of the findings you discuss.

Response: We have integrated the quotes into the text to follow the description of the relevant findings.

The paper is lacking in analysis and insight. The conclusion, that incarcerated women should be given access to reproductive health care, is true but not unique and not directly tied to most of the findings that the authors present.

Response: We have modified our conclusion to emphasize both that women experiencing imprisonment should have better access to healthcare, but also that discrimination and stigma felt by women must be addressed in order to promote reproductive justice for women in prison. We believe this conclusion is more closely tied to our findings, and that it makes the main message of our paper more unique.

Reviewer #3: The article is an important contribution to the literature on incarcerated women's health and access to reproductive healthcare during imprisonment. There are two areas where this article can be strengthened and improved. One, the authors neglect to cite literature on family planning for incarcerated and formerly incarcerated women. Doing so would further substantiate some of the findings in this study. 

Response: We have cited the following papers on family planning for women who experience imprisonment:

Clarke JG, Hebert MR, Rosengard C, Rose JS, DaSilva KM, Stein MD. Reproductive health care and family planning needs among incarcerated women. Am J Public Health 2006; 96(5): 834-9.

Peart MS, Knittel AK. Contraception need and available services among incarcerated women in the United States: a systematic review. Contracept Reprod Med 2020; 5: 2.

Sufrin CB, Creinin MD, Chang JC. Contraception services for incarcerated women: a national survey of correctional health providers. Contraception 2009; 80(6): 561-5.

Schonberg D, Bennett AH, Sufrin C, Karasz A, Gold M. What Women Want: A Qualitative Study of Contraception in Jail. Am J Public Health 2015; 105(11): 2269-74.

Schonberg D, Bennett AH, Gold M. The contraceptive needs and pregnancy desires of women after incarceration: A qualitative study. Contraception 2020; 101(3): 194-8.

Second, the authors should also address potential ethical issues that are relevant when conducting studies in prison and how they addressed those ethical issues in their work. For example, receiving IRB approval is the bare minimum standard. What ethical issues did they consider prior to launching their work and how did they resolve them to proceed with this work.

Response: We have included a more detailed description of our study procedures, which outlines the issues we faced in proceeding with this work, under ‘Study Development’ (lines 94-121).

---

## [Decision Letter · Decision Letter 1]

18 Mar 2021

PONE-D-20-34245R1

Reproductive healthcare in prison: A qualitative study of women’s experiences and perspectives in Ontario, Canada

PLOS ONE

Dear Dr. Liauw,

Thank you for submitting your revised manuscript to PLOS ONE. After careful consideration, we feel that it has merit but does not fully meet PLOS ONE’s publication criteria as it currently stands. Therefore, we invite you to submit an additional revised version of the manuscript that addresses the points raised during the review process.

I am in agreement with the reviewers that the revisions made to the manuscript have substantially improved it. There remain several small issues, as outlined in the reviewer comments below, that should be addressed prior to acceptance. Overall, however, I am enthusiastic about this manuscript.

We look forward to receiving your revised manuscript.

Kind regards,

Andrea Knittel

Academic Editor

PLOS ONE

Journal Requirements:

Reviewers' comments:

Reviewer's Responses to Questions

**Comments to the Author**

1. If the authors have adequately addressed your comments raised in a previous round of review and you feel that this manuscript is now acceptable for publication, you may indicate that here to bypass the “Comments to the Author” section, enter your conflict of interest statement in the “Confidential to Editor” section, and submit your "Accept" recommendation.

Reviewer #1: All comments have been addressed

Reviewer #2: (No Response)

Reviewer #3: All comments have been addressed

2. Is the manuscript technically sound, and do the data support the conclusions?

Reviewer #1: Yes

Reviewer #2: Yes

Reviewer #3: Yes

3. Has the statistical analysis been performed appropriately and rigorously? 

Reviewer #1: N/A

Reviewer #2: N/A

Reviewer #3: N/A

4. Have the authors made all data underlying the findings in their manuscript fully available?

Reviewer #1: Yes

Reviewer #2: Yes

Reviewer #3: Yes

5. Is the manuscript presented in an intelligible fashion and written in standard English?

Reviewer #1: Yes

Reviewer #2: Yes

Reviewer #3: Yes

6. Review Comments to the Author

Reviewer #1: Really lovely revision. I have two minor comments. The first is to ask for a bit of clarification about the topic of accessing contraception while incarcerated. Readers may wonder why is this necessary--you have provided one strong rationale about continuity (starting while incarcerated will make it easier to continue when out), but some additional text on this would be helpful. Also, why women would need condoms while locked up needs some explanation.

Second, I find the role of correctional officers as gatekeepers to health services horrifying and I would suggest that the topic of reproductive health care be considered under the umbrella of overall health care. A final plea/recommendation might be along the lines that women are whole persons deserving of health care, including reproductive health services. While the focus is reproductive health, we want all of the health services for women to be improved.

Once again, nice job!

Reviewer #2: I appreciate the effort the authors put into revising the paper and it is greatly improved as a result. I have a few remaining, fairly minor comments:

Background: Situate the literature cited. Is it from North America? Western Countries? Worldwide?

Approach lines 68-71: Unclear what the authors means when they say they did not 'assume a pre-specified framework' in order to 'build off previous work in this area.' Are they referring to the authors’ previous work or the literature? It seems like building off previous work would contribute to specifying a theoretical framework. Clarify.

Study development section: While it’s helpful for the reviewers who saw the first draft of the paper, I don’t it's necessary to include the information about the community sample, since you are not using the data. The summary of quantitative results is a good addition.

Line 134: Where were flyers posted (common areas? restrooms?) and announcements made?

Lines 139-146: Too much detail about the facilitator. I would restrict the description of the facilitator to say they had training in qualitative methods and feminist epistemologies.

Line 150: Were any 'ground rules' discussed for focus groups? (eg, confidentiality among participants)

Lines 166-168: The reference to the previous set of reviews is likely to be confusing to a new reader.

Lines 168-172: Most of this belongs in the results section. Here, I would simply say that you used a Reproductive Justice Framework with a citation.

Line 180: How many women were in each focus group?

Lines 188-193: The discussion of subcategories is confusing, in terms of which belong to which larger categories. It may be clearer to simply refer to the figure.

Line 200: Mention why this subcategory was not included in the final analysis

The reorganization of quotes to support areas of analysis has strengthened the paper tremendously. They now have a lot more impact

Line 415: quote a little confusing. Who are 'they'? Establish context.

Discussion: I would love to see a paragraph highlighting the excellent and concrete ideas participants shared for improving reproductive health care in the prison setting. I highlighted a few as I read:

contraception should be discussed during routine release planning

contraception should be discussed and offered during the routine nursing assessment on prison admission

staff should ask women about their current needs for and interest in contraception

I would also like to see the issue of miscarriage highlighted a bit in the discussion, as it seemed important to women's experiences and simple interventions like providing sufficient access to menstrual pads could reduce misery and stigma

Reviewer #3: It appears that the authors have adequately responded to the reviewers concerns. I think this article will make a meaningful contribution to the literature.

7. PLOS authors have the option to publish the peer review history of their article (what does this mean?). If published, this will include your full peer review and any attached files.

Reviewer #1: **Yes: **Patricia J Kelly

Reviewer #2: No

Reviewer #3: No

---

## [Author Response · Author response to Decision Letter 1]

27 Apr 2021

Reviewer #1: Really lovely revision. I have two minor comments. The first is to ask for a bit of clarification about the topic of accessing contraception while incarcerated. Readers may wonder why is this necessary--you have provided one strong rationale about continuity (starting while incarcerated will make it easier to continue when out), but some additional text on this would be helpful. Also, why women would need condoms while locked up needs some explanation.

Response: Thank you for this feedback. We have added the following content to the Background: 

“The World Health Organization and the United Nations recommend the provision of reproductive healthcare, specifically including contraception and pregnancy-related care, for women in prison [14]. Access to contraception in prison can support women in preventing unintended pregnancy after release, which is particularly important since women may be at increased risk of unintended pregnancy during that period and they often face barriers to contraception access in the community, including urgent competing priorities while transitioning back to the community [15]. Contraception access in prison is also important for females who are sexually active with males in prison.” (Lines 54-61).

References 14 and 15 were added to support these points. 

Second, I find the role of correctional officers as gatekeepers to health services horrifying and I would suggest that the topic of reproductive health care be considered under the umbrella of overall health care. A final plea/recommendation might be along the lines that women are whole persons deserving of health care, including reproductive health services. While the focus is reproductive health, we want all of the health services for women to be improved.

Once again, nice job!

Response: Thank you, we agree with highlighting the importance of improving health services for women overall. We have added the following italicized content to the discussion: 

“Our results also raise the question, as posed by other reproductive justice scholars [40], as to whether reproductive justice can in fact be achieved in prison since these data suggest that women’s desires to have better quality and access to reproductive healthcare, and healthcare overall, has largely been negatively impacted by the conditions of their arrest and incarceration. Ultimately, although our study was focused on reproductive healthcare, our findings highlight ways in which systemic changes are necessary to promote the overall health of women in prison.” (Lines 518-524).

Reviewer #2: I appreciate the effort the authors put into revising the paper and it is greatly improved as a result. I have a few remaining, fairly minor comments:

Background: Situate the literature cited. Is it from North America? Western Countries? Worldwide?

Response: We have specified where the cited literature is situated throughout the Background. Most of the literature is from North America, although some references are based on research in other areas. We also removed one reference (Covington, S.S., Women and the criminal justice system. Womens Health Issues, 2007. 17(4): p. 180-2.) since it was an editorial without original data. 

Approach lines 68-71: Unclear what the authors means when they say they did not 'assume a pre-specified framework' in order to 'build off previous work in this area.' Are they referring to the authors’ previous work or the literature? It seems like building off previous work would contribute to specifying a theoretical framework. Clarify.

Response: We had originally meant building off our previous work in this area, but on reviewing this line again we agree with reviewer that it was unclear, so we removed “to build off previous work in this area” from this section (Lines 77-79).

Study development section: While it’s helpful for the reviewers who saw the first draft of the paper, I don’t it's necessary to include the information about the community sample, since you are not using the data. The summary of quantitative results is a good addition.

Response: We agree with this suggestion, and have removed the information about the community sample (Lines 120-123).

Line 134: Where were flyers posted (common areas? restrooms?) and announcements made?

Response: We have added that flyers and announcements were made in common areas in the prison (Line 136-137).

Lines 139-146: Too much detail about the facilitator. I would restrict the description of the facilitator to say they had training in qualitative methods and feminist epistemologies.

Response: We have shortened the description as suggested: 

“The focus groups were facilitated by one female team member (JJ), who had graduate-level training in qualitative methods and feminist epistemologies.” (Lines 142-143).

Line 150: Were any 'ground rules' discussed for focus groups? (eg, confidentiality among participants)

Response: We have added the following details (italicized below) about the ground rules discussed during the focus groups:

“In the letter of information participants were informed that they were able to opt out of the study at any time, and that participation in the study would not impact their treatment in the facility. They were asked to be respectful of each other’s experiences given the sensitive and personal nature of the conversations and to keep what was shared in the group confidential and were also reminded that confidentiality could not be guaranteed following the group. The facilitator reviewed the letter of information and consent form verbally and also made participants aware of her affiliation, research background, context of the current research as situated in the previous research completed on the topic by the researchers, and the objectives of the research.” (Lines 146-154).

Lines 166-168: The reference to the previous set of reviews is likely to be confusing to a new reader.

Response: We agree with this feedback and have simplified this section to the following: 

“After initial review of our results, we considered using a reproductive justice framework to enhance our analysis and interpretation” (Lines 170-171).

Lines 168-172: Most of this belongs in the results section. Here, I would simply say that you used a Reproductive Justice Framework with a citation.

Response: Thank you for this feedback. We agree and have moved this section to Results (Lines 188-192). We changed this to say we used a reproductive justice framework, with a citation as suggested (Lines 170-171).

Line 180: How many women were in each focus group?

Response: We have clarified the number of women in each group, by modifying this sentence to the following:

“We conducted three focus groups with seven women in each group, for a total of 21 participants in a provincial prison.” (Line 179).

Lines 188-193: The discussion of subcategories is confusing, in terms of which belong to which larger categories. It may be clearer to simply refer to the figure.

Response: We have made the suggested changes: 

“Within these categories, the data suggested four sub-categories (see Figure 1).” (Line 187-188).

Line 200: Mention why this subcategory was not included in the final analysis

Response: We have added the following to our caption for Figure 1, and this is also explained in the Results section (Lines 196-198).

“Sub-category ‘challenges with contraception’ was not included in our final analysis in order to streamline results, as the relevant concepts within this sub-category were captured within the other categories.” (Lines 245-247).

The reorganization of quotes to support areas of analysis has strengthened the paper tremendously. They now have a lot more impact

Line 415: quote a little confusing. Who are 'they'? Establish context.

Response: We clarified that ‘they’ referred to correctional officers (Line 420).

Discussion: I would love to see a paragraph highlighting the excellent and concrete ideas participants shared for improving reproductive health care in the prison setting. I highlighted a few as I read:

contraception should be discussed during routine release planning

contraception should be discussed and offered during the routine nursing assessment on prison admission

staff should ask women about their current needs for and interest in contraception

Response: Thank you for this suggestion. We added the following to the Discussion:

“Participants had several suggestions to improve reproductive healthcare in prison, including making information regarding contraceptive options more readily available prior to seeing a doctor, asking women about contraceptive preferences on admission to the prison, and improving continuity with services in the community to access contraception.” (Lines 529-533).

I would also like to see the issue of miscarriage highlighted a bit in the discussion, as it seemed important to women's experiences and simple interventions like providing sufficient access to menstrual pads could reduce misery and stigma

Response: We agree that pregnancy loss was an important issue to highlight. We have added the following text to the Discussion:

“In particular, with respect to pregnancy, participants shared how lack of access to adequate healthcare and health resources contributed to trauma surrounding pregnancy loss or the potential for pregnancy loss while being detained.” (Lines 453-455).

Reviewer #3: It appears that the authors have adequately responded to the reviewers concerns. I think this article will make a meaningful contribution to the literature.

Response: Thank you!

---

## [Editor Report · Decision Letter 2]

5 May 2021

Reproductive healthcare in prison: A qualitative study of women’s experiences and perspectives in Ontario, Canada

PONE-D-20-34245R2

Dear Dr. Liauw,

We’re pleased to inform you that your manuscript has been judged scientifically suitable for publication and will be formally accepted for publication once it meets all outstanding technical requirements.

Kind regards,

Andrea Knittel

Academic Editor

PLOS ONE
---

## [Editor Report · Acceptance letter]

10 May 2021

PONE-D-20-34245R2 

Reproductive healthcare in prison: A qualitative study of women’s experiences and perspectives in Ontario, Canada 

Dear Dr. Liauw:

I'm pleased to inform you that your manuscript has been deemed suitable for publication in PLOS ONE. Congratulations! Your manuscript is now with our production department. 

Kind regards, 

on behalf of

Dr. Andrea Knittel 

Academic Editor

PLOS ONE